# Corrosion Activity of Stainless Steel SS430 and Carbon Steel B450C in a Sodium Silicate Modified Limestone-Portland Cement Extract

**DOI:** 10.3390/ma16145066

**Published:** 2023-07-18

**Authors:** David Bonfil, Lucien Veleva, Sebastian Feliu, José Iván Escalante-García

**Affiliations:** 1Center for Research and Advances Study (CINVESTAV), Applied Physics Department, Campus Merida, Merida 97310, Mexico; david.bonfil@cinvestav.mx; 2National Center for Metallurgical Research (CENIM-CSIC), Surface Engineering Corrosion and Durability Department, 8040 Madrid, Spain; sfeliu@cenim.scic.es; 3Center for Research and Advanced Study (CINVESTAV), Campus Saltillo, Ramos Arizpe 25900, Mexico; ivan.escalante@cinvestav.edu.mx

**Keywords:** corrosion, carbon steel, stainless steel, alkali-activated cement, Portland cement, cement extract solution, corrosion potential, pH, SEM-EDS, XPS, EIS

## Abstract

Stainless steel SS430 and carbon steel B450C were exposed for 30 days to the aqueous extract of sodium silicate-modified limestone-Portland cement as an alternative for the partial replacement of the Portland cement clinker. The initial pH of 12.60 was lowered and maintained at an average of 9.60, associated with air CO_2_ dissolution and acidification. As a result, the carbon steel lost its passive state, and the corrosion potential (OCP) reached a negative value of up to 296 mV, forming the corrosion layer of FeO, and FeOOH. In the meaning time, on the stainless steel SS430 surface, a passive layer of Cr_2_O_3_ grew in the presence of FeO, Fe_2_O_3_ and Cr(OH)_3_ corrosion products; thus, the OCP shifted to more positive values of +150 mV. It is suggested that a self-repassivation process took place on the SS430 surface due to the accumulation of alkaline sulfates on the interface. Because of the chloride attack, SS430 presented isolated pits, while on B450C, their area was extended. The quantitative analysis of EIS Nyquist and Bode diagrams revealed that the Rp of the corrosion process for SS430 was 2500 kΩcm^2^, ≈32 times lower in magnitude than on B450C, for which the passive layer tended to disappear, while that on SS430 was ≈0.82 nm.

## 1. Introduction

The global Portland cement (PC) production was 4.1 billion tons in 2021 [1], with estimated emissions of 0.87 tons of CO_2_ per 1 ton of PC [2], contributing 8–10% of the anthropogenic CO_2_ global emissions [3]; these figures could increase due to the growing global urbanization. To mitigate the negative environmental impact of the above, several alternative types of cement have been proposed. The two of the most common alternative types of cement include: (a) blended cement, in which PC is partially replaced by one or more supplementary cementitious materials (limestone, blast furnace slag, fly ash, calcinated clays, natural pozzolan, waste glass, etc.) [4,5,6,7,8,9,10]; and (b) alkali-activated cement (AAC), based on the reaction of a precursor (i.e., the supplementary cementitious materials referred above, limestone, etc.) and alkaline activators (alkali silicates and/or hydroxides) [11]. The CO_2_ emission during the manufacture of these alternative types of cement could be up to 5–6 times lower relative to PC [12] in addition to some of the features, such as good mechanical strength, a dense pore structure [13], thermal resistance [14,15], etc.

Limestone (LS) and waste glass have been used as partial replacements for PC without compromising the mechanical properties [16,17] and as precursors in AAC [18,19]. Limestone-based AAC have been reported to develop compressive strength by the formation of C-S-H and N-C-S-N hydration products intermixed with carbonates and silica gel N-S-H (15–25 MPa after 360 days) [18], while waste glass-based AAC have been reported to form C-S-H intermixed with a silica gel (15–29 MPa after 28 and 180 days) [19]. Investigations of AAC in waste glass incorporating limestone as a precursor and as an aggregate [20] have indicated that the limestone participated physically and chemically in the reaction process, densifying the microstructure and enhancing its mechanical properties and stability underwater (15–20 MPa at 90 days). On the other hand, based on the above-blended types of cement, PC-limestone has been modified with an alkaline activator synthesized from waste glass, resulting in a novel alternative “green” cement, which could reach 50 MPa in compressive strength after 90 days [21].

In addition to the strength requirements, the application of new cement requires the characterization of the electrochemical behavior of the steel reinforcement due to the internal evolution of the chemical environment of the concrete. The pH of the pore solution given by the content of alkalis in the cement and the saturation with Ca(OH)_2_ is a key factor for promoting the passivation of reinforcement steels (pH > 13), leading to the formation of a thin layer of Fe-oxides/hydroxides on the surface of carbon steel known as the passive layer, which protects the steel surface from the advancement of corrosion in the presence of aggressive corrosive agents [22,23,24]; for stainless steel, the passive layer is primarily composed by a Cr-rich oxide layer that features self-generation [25]. It is reported that the very high alkalinity of AAC could be favorable for the corrosion protection of the embedded reinforcing steel [26]; some studies have reported a pH ≈ 13 of the aqueous pore solution [27,28,29], which is similar or higher than that of the PC extract solution.

To circumvent the experimental difficulties of this study on the corrosion behavior of steel embedded in concrete (electrode and cell designs, position of reference and auxiliary electrodes, drop potential IR in concrete and its compensation, restriction of oxygen diffusion, the development of macro-corrosion cells, among others), the model solutions were proposed to obtain comparative results while maintaining the control of some parameters, which is difficult to accomplish in reinforced-concrete samples. The influence of the ionic composition of the cement pore solution on the steel passivation behavior was investigated using aqueous saturated Ca(OH)_2_ (pH 12–13) [30,31,32], KOH and NaOH solutions [33,34,35,36] as models, and cement extract (CE) solutions with a variety of ions (Ca^2+^, Na^+^, K^+^, OH^−^, and SO_4_^2−^) [37,38,39,40]. For example, it is considered that the Ca^2+^ ion could affect the composition and the protective efficiency of the passive layer [41], while silicates could promote steel passivation and its resistance to a chloride attack [42]. These results suggested that the electrochemical behavior of low carbon steel was very similar to NaOH and saturated Ca(OH)_2_ model solutions, while in a PC extract solution, the charge transfer resistance to the corrosion process increased due to the adsorption of C-S-H gel on the steel surface [43]. Mild steel rebars have been exposed to alkali NaOH solutions to simulate the chemistry of the AAC concrete pore solution, and in the presence of chloride ions, the depassivation of the steel surface was found to be strongly dependent on the pH pore alkalinity [44]. On the other hand, EIS electrochemical measurements have shown that the extract of AAC based on furnace slag had a great capacity to passivate the carbon steel HRB400 surface and offered protective efficiency against chlorides [45]. After the exposure of carbon steel HPD235 to an extract solution of a highly alkaline AAC based on fly ash, the passive layer was composed of an inner layer of FeO and an outer of FeOOH/Fe_2_O_3_. The presence of S-species in the concrete pore solution of AAC led to the formation of a passive layer that was more susceptible to the attack of chloride ions, and the corrosion resistance of the passive layer was reduced [46,47,48,49,50]. Conversely, another study reported that the deposition of S, due to the oxidation of sulfide anions to sulfur in the pore solution of a slag-based AAC, prevented potential rising into the pitting regimen and led to the inhibition of the oxygen cathode reaction in mild steel [51].

In our previous work [52], the electrochemical behavior of carbon steel B450C and low-chromium ferritic stainless steel SS430 was characterized after their exposure for up to 30 days to the CE solution (pH = 12.38) of a “green” super sulfated cement, which could be considered as an alternative to PC, based on 52% pumice (SiO_2_/Al_2_O_3_), 34% CaSO_4_·½H_2_O as a sulphatic activator, 7% of Portland cement (PC) and CaO as alkaline activators. The initial pH of 12.38 dropped from the first day to 7.84, accompanied by a shift in the free corrosion potential (OPC) of the carbon steel up to −480 mV negative values, losing its passive state and leading to the formation of corrosion products α-, γ-FeOOH and Fe_2_O_3_; meanwhile, the stainless steel SS430 preserved its passive state at a positive OCP (+182.50 mV) due to the formation of a thin layer of ~0.8 nm Cr_2_O_3_ [46]. On both surfaces, a localized corrosion attack was observed, which was influenced by the presence of Cl-ions in the CE solution originating from the pumice.

In this study, martensitic carbon steel B450C and low-chromium ferritic stainless steel SS430 were exposed for up to 30 days to the aqueous extract of a sodium silicate-modified limestone-Portland cement named “JLSC1” [21], which was composed of 30% PC, 23.3% sodium silicate activator, 26.5% LS and 0.2% superplasticizer, to simulate the environment at the steel–concrete pore interface. The pH of the CE solution and the change in time of the steel-free corrosion potential (OCP) were monitored during the immersion test. Electrochemical impedance spectroscopy (EIS) measurements were performed to obtain the parameters of the interface steel-CE pore solution. The steel surfaces were characterized by the SEM-EDS, XPS techniques. To the best of our knowledge, no other research on this topic has been previously undertaken.

## 2. Materials and Methods

### 2.1. Steel Samples and Surface Characterization

Flat samples of low chromium ferritic stainless steel SS430 (Outokumpu, Espoo, Finland) and martensitic carbon steel B450C (Pittini Group, Gemona del Friuli, Italy) were cut (0.8 cm^2^), abraded with SiC paper to 4000 grit using ethanol as a lubricant, then sonicated for 10 min (Branson 1510, Branson Ultrasonics Co., Danbury, CT, USA) and dried at room temperature (294 K or 21 °C). The steel nominal composition (wt.%) is presented in Table 1, according to the manufacturers. 

According to [40], SEM-EDS analysis identified black dots on the carbon steel surface, which contained 5.02% C, 1.31% Mn, and 0.38% S, ascribed to the phases of MnS and Mn3C [53]. Due to the quality of the scrap, Cu (0.83%) was also detected. On the stainless steel surface, zones with 24.03% Cr, 30.04% C, and 3.27% N were detected and associated with chromium nitride and carbide (Cr,Fe)_7_C_3_ [54,55]. The presence of 0.7% of V was ascribed to precipitates of V6C5 and VB-nitride phases. The presence of Si and C, associated with the SiC phases, was detected on both steel surfaces [53].

### 2.2. Sodium Silicate Modified Limestone-Portland Cement “JLSC1” and Its Extract Solution

The cement used in this research, labeled “JLSC1”, is that described in [21], which details that its manufacture has a 50% lower global warming potential (kg CO_2_-eq/m^3^) compared to a PC which can reach 50 MPa at 90 days. Table 2 shows the oxide composition of the JLSC1 cement.

The main oxides in JLSC1 (Table 2) are CaO and SiO_2_; however, the content of CaO (47.53%) is lower than that of Portland cement (66.84–58.4% CaO) because the JLSC1 contains only 30% of PC; the SiO_2_ content (22.80%) is comparable to that of PC (21.35–22.30%) and can be attributed to the sodium silicate activator (23.3% in JLSC1). Other oxides present at a low content included 3.25% Na_2_O, 1.72% Al_2_O_3_, 1.13% SO_3_, 1.11 Fe_2_O_3_, 0.61% MgO, and 0.38% K_2_O.

The aqueous cement extract was prepared from a 1:1 wt./wt. mixture of JLSC1 cement and ultrapure deionized water (18.2 MΩ cm). The mixture was agitated and left for 24 h to hydrate in a closed container. Then, the supernatant was filtered with 2.5 µm pore size filter paper (Whatman, Kent, UK) to remove the particles and was kept in a sealed container. Table 3 presents the chemical composition of the JLSC1 cement extract solution, characterized by absorption spectrometry and atomic emission by plasma; the Cl^−^ ion content was determined by the ion-selective electrode.

The low Ca^2+^ content could be attributed to the consumption of these ions for the formation of hydrated calcium silicates (C-S-H). On the other hand, the high Na^+^ content (2700 mg/L) came from the silicate activator (powdered sodium waste glass, PSWG) after reacting with limestone and portlandite (Ca(OH)_2_) when the Na fraction reacted with water to form a NaOH buffer for the alkalinity of the JLSC1-CE solution. Likewise, the K^+^ content (1668.4 mg/L) was a part of K_2_O’s hydration, where the KOH product contributed to the alkalinity of the JLSC1-CE solution (pH ≈ 12.60). Meanwhile, the 1132 mg/L SO_4_^2−^ content was attributed to the hydration of the SO_3_ fraction, which could react with the hydroxides resulting in the formation of sulfates [56].

### 2.3. Immersion Test

Triplicated steel samples (0.8 cm^2^) were immersed in 10 mL of a JLSC1 cement extract solution for a period of 720 h (30 days), in sealed containers (with paraffin tape), according to the standard ASTM-NACE/ASTM G31-12a [57]. After 168 h (7 days) and 720 h (30 days), the samples were withdrawn, rinsed with deionized water, and dried in the air at room temperature (294 K or 21 °C). The pH of the cement extract solution was measured (PH60 Premium Line, pH tester, Apera Instruments, LLC, Columbus, OH, USA) after the withdrawal of steel samples at each period of exposure. The surface of the samples after the corrosion immersion tests were characterized by scanning electron microscopy (SEM-EDS, XL–30 ESEM-JEOL JSM-7600F, JEOL Ltd., Tokyo, Japan). The corrosion products were analyzed with X-ray photoelectron spectroscopy (XPS, K-Alpha, Thermo Scientific, Waltham, MA, USA) after sputtering the surface for different times (seconds) with a scanning Ar-ion gun. The formed layers were removed [58], and the surfaces were characterized by SEM-EDS and XPS.

### 2.4. Electrochemical Measurements

A typical three-electrode cell configuration (inside a Faraday cage), connected to a potentiostat (Interface-1000E potentiostat/galvanostat/ZRA, Gamry Instruments, Philadelphia, PA, USA), was used for the electrochemical experiments (294 K or 21 °C): the steel plates of B450C and SS430 were the working electrodes, the Pt plate was an auxiliary, and a saturated calomel electrode (SCE) was used as the reference electrode. During the electrochemical experiments, the change in time of the open circuit potential (OCP) was monitored and considered for the free corrosion potential of the studied steels. Electrochemical impedance spectroscopy (EIS) at open circuit potential (OCP) was performed, applying an alternating current (AC) signal of ±10 mV in amplitude at a frequency range from 100 kHz to 10 mHz and with a sampling size of 10 data points/decade. Nyquist and Bode EIS diagrams were recorded at different immersion periods: 1, 7, 14, 21, and 30 days. The data were analyzed with Gamry Echem Analyst^®^ (version 7.1, Philadelphia, PA, USA).

## 3. Results

### 3.1. Change in Time of pH of JLSC1 Cement Extract Solution and Corrosion Potential (OCP) of the Steels up to 30 Days of Immersion 

Figure 1 presents the change in time for the pH of the JLSC1-CE solution and the corrosion potential (OCP) during the exposure for up to 30 days of SS430 (Figure 1a) and B450C (Figure 1b) steels. The initial pH was 12.60, compared to that of the Portland cement extract (pH = 13) [40]. 

On the first day of exposure (24 h) of B450C and SS430, the pH of JLSC1 tended to lower the values between 10.50 and 10.92, which reached an average value of 9.68 after 7 days and almost maintained an ending at 9.60 after 30 days of immersion (Figure 1). Meanwhile, the OCP of stainless steel SS430 (−199.96 mV) showed a tendency to more positive values, indicating the development of a more protective passive layer along the time (≈+155 mV); a similar tendency of OCP was observed for SS430 in the PC-extract solution (≈+217 mV) at 30 days of immersion [40]. On the other hand, the OCP of carbon steel B450C (−156.30 mV) tended to offer more negative values from the first 24 h (−296.58 mV) and ended at a less negative OCP around −181.50 mV due to the formed corrosion layer, which acted as a physical barrier. Comparing the behavior of the studied steels exposed to the JLSC1 extract to that in the PC extract solution [40], the OCP values of B450C carbon steel were 1.8 times greater (more negative) since the first 24 h, indicating the loss of the passive state.

The change in time of the corrosion potential values (OCP) strongly depended on the change in time of the pH in the extract solution. A decrease in pH could be associated with the dissolution of the CO_2_ from the air environment, which, combined with water forming on the carbonic acid (H_2_CO_3_), could later dissociate into bicarbonate ions (HCO_3_^−^) and hydrogen ions (H^+^), promoting the acidification of the 10 mL extract solution used in this study for the steel immersion test [59]. It was considered that for 7 < pH < 10, the HCO_3_^−^ ions were the predominant species, very corrosive to metal, and the release of Fe^2+^ occurred, forming iron hydroxide II (consuming the OH^−^ ions and lowering the pH) [60]. However, the established dynamic equilibrium between the formed weak carbonic acid and the bicarbonate ions (as its conjugated base) of such a formed system could act as a buffer for the pH change. On the other hand, the diminishing pH of the JLSC1 extract solution should also be considered due to the lower amount of portlandite Ca(OH)_2_ in the concrete pore solution and, thus, it had a lower ability to maintain the pH [61]. 

### 3.2. B450C Carbon Steel Surface Characterization after Exposure to JLSC1 Cement Extract Solution 

Figure 2 shows SEM images of the carbon steel surfaces, and Table 4 presents the EDS analysis (wt.%) after 7 and 30 days of exposure to the JLSC1 cement extract solution. After 7 days of exposure (Figure 2a) at pH of 9.77 and OCP = −259.97 mV, the steel surface was not completely covered by a corrosion layer; zone A presented the matrix composition of the steel: Fe (88.71%), C (5.00%), Mn (0.74%) with some traces of elements, such as Na, K, and Si. For zones B and C, the EDS analysis reported a greater content of Na, O, S, and K, which could be attributed mainly to their sulfates as a part of the cement extract (Table 3), and Ca-carbonate in a lower content.

At 30 days of exposure to the carbon steel of the JLSC1 extract solution (Figure 2b), the surface was covered by a corrosion layer enriched in Fe and O (Table 4) and was associated with the formation of Fe-oxides and hydroxides of different morphology and accompanied with some traces of S, Si, Mn, Cl. 

The XPS spectra of the carbon steel B450C, after 7 and 30 days of exposure to the JLSC1 cement extract solution, are present in Figure 3 and Figure 4, respectively. The peak of the Fe2p (Figure 3a) spectra was deconvoluted and associated with Fe_2_O_3_/FeOOH (711.1–711.6 eV) and FeO (709.8 eV), as a part of the passive layer, and Fe metal as a part of the steel matrix (706.8 eV) [62]. The peak of O1s (Figure 3b) were deconvoluted in three peaks, which were ascribed to oxides O^2−^ (531.3 eV), OH^−^ (532.8 eV), and C-O (532.17 eV). The intensity of the hydroxide peak was 3.5 times lower than that of the oxides, indicating that Fe_2_O_3_ and FeO oxides prevailed with respect to FeOOH; with the increase in the sputtering time, the intensity of O1s diminished due to the depletion of oxides and hydroxide species. The peaks of C1s and S2p were attributed to carbonate and sulfate crystals, which was suggested by the SEM-EDS analysis (Figure 2, Table 4) [63,64]. At 30 days of exposure (Figure 4), when the carbon steel formed a thicker corrosion layer (Figure 2b), the identified peaks were Fe2p and O1s. The deconvolution of Fe2p (Figure 4a,b) suggested the presence of FeOOH (711.48 eV) and FeO (710.0 eV) [62] in a similar proportion, according to the intensity of O1s (Figure 4d).

### 3.3. SS430 Stainelss Steel Surface Characterization after Exposure to JLSC1 Cement Extract Solution 

Figure 5 presents SEM images of stainless steel SS430 after exposure to the JLSC1 extract for 7 days (Figure 5a) and 30 days (Figure 5b). The EDS analysis of some zones of interest is resumed in Table 5. Although the pH of the cement extract solution diminished to 9.58, after 7 days of exposure, the steel corrosion potential of OCP was positive (+128 mV), as an indication that the metal surface maintained its passive state (Figure 1). According to EDS analysis (Table 5), zone A presented O (36.20%), S (22.87%), K (31.19%) and Na (8.28%), which could be associated with the precipitation of sulfates K, and Na (Table 3). Zone B corresponded to the steel matrix. In zone C, the precipitation of sulfate crystals was observed and composed of Na (21.69%), O (14.62%), S (4.77%), and in the lower content of C (1.88%). 

After 30 days of exposure to the JLSC1 cement extract (Figure 5b), the surface of SS430 maintained its passive state (Figure 1, OCP of +154.73 mV), and it was not fully covered by a corrosion layer. The high content of O, K, S, and Na in zone 1 (Table 5) could be attributed to the precipitation of K and Na sulfates (Table 3). Meanwhile, the EDS of zone 2 indicated the higher contents of C (15.31–17.02%) and Fe, which were associated with the (Cr,Fe)_7_C_3_ carbide [54,55], and were reported for the reference samples of this steel.

It has been suggested that anions with a higher charge (SO_4_^2−^, 1132.0 mg/L, Table 3) may accumulate at the metal–solution interface inside the pits and lead to a reduction effect on the electrical potential and, thus, to the self-repassivation of the steel surface [65].

The XPS spectra of stainless steel SS430, after exposure for 30 days to the JLSC1 cement extract solution, are presented in Figure 6. Based on the deconvolution of Fe2p (Figure 6a), Cr2p (Figure 6b), and O1s (Figure 6c) peaks, it could suggest that the passive layer formed on the stainless steel surface was composed of FeO (709.1 eV), Fe_2_O_3_ (711.1 eV), Cr_2_O_3_ (576.5 eV), and the corrosion products of Cr(OH)_3_ [62,66]. The deconvolution of the O1s confirmed the peaks of the oxides (531.3 eV) and oxide-hydroxide OH^−^ (533.1 eV), and in a low intensity, the peak of C-O (535.8 eV) could be associated with the presence of Na (Na1s) and K (K2p) carbonates or bicarbonates; the peaks of Cl2p, C1s, S2p are also indicated. 

### 3.4. Steel Surface Damage after the Exposure to JLSC1 Cement Extract Solution 

Figure 7 compares SEM images of stainless steel SS430 and carbon steel B450C surfaces after the removal of layers formed during their exposure to the JLSC1 cement extract for 30 days. On the stainless-steel surface (Figure 7a), isolated pits (red-colored circles) are visible with a diameter lower than 1–3 µm due to the presence of chloride ions in the cement extract (88 mg/L, Table 3); these ions penetrated the oxide passive film. These localized attacks could appear in the vicinity of particles with cathodic activity. Such particles (named A, B, and C) were those suggested by EDS analysis (Table 6), presenting high contents of Fe, Cr, and C and a lower content of V, which could be ascribed mainly to Cr-Fe carbides (Cr,Fe)_7_C_3_ [54,55] and the precipitates of V_6_C_5_ phase, all characteristic of the SS430 matrix (reference samples). On the carbon steel surface (Figure 7b), the attacks were more uniform because this steel lost its passive state due to a shift in the JLSC1 cement extract pH to lower alkaline values. Table 7 resumes the EDS analysis of the zones of interest. The content of Mn could be ascribed to the phases of MnS and Mn3C [53]. 

### 3.5. Electrochemical Impedance Spectroscopy (EIS) Diagrams

Figure 8 compares the Nyquist impedance diagrams of stainless steel SS430 (a) and carbon steel B450C (b) when exposed for 30 days to the JLSC1 extract solution. These diagrams of stainless steel SS430 (Figure 8a) at 1 day and 7 days were almost similar, displaying a linear-diffusion impedance (at the low-frequency domain, 10–100 mHz) and indicating that a passive film was formed on the steel surface; this controlled the oxygen diffusion and mass transfer. From 7 days of steel exposure to the JLSC1 cement extract solution, a semi-linear diffusion impedance was observed due to the thickening of the passive layer, which was mainly composed of Cr_2_O_3_ with the addition of Fe_2_O_3_ and FeO oxides; this fact corroborates with the shift in the corrosion potential (OCP) to more positive values of ≈+142 mV (Figure 1) at that time of exposure. Towards the end of the experiment, after 30 days, the shape of the Nyquist diagrams presented a tendency to form an incomplete depressed semi-circle, which could be associated with the events of localized pitting attacks due to the ingress of chloride ions through the pores of the formed film. Therefore, at 10 mHz, the imaginary value of impedance Z″ decreased (14.7 kΩ cm^2^), while the value of the real part Z′ increased (72.8 kΩ cm^2^).

On the other hand, the Nyquist diagram of carbon steel B450C (Figure 8b), after 1 day of exposure to the JLSC1 cement extract, showed an incomplete semi-circle, with low Z′ and Z″, due to a drop in the pH to 10.90 and the loss of its passive state, which corroborated with the shift of the corrosion potential (OCP) to more negative values (−296 mV), indicating progress in the corrosion process. Since the thickness of the corrosion layer increased over time (precipitation of sulfates and carbonates were revealed by SEM-EDS analysis, besides the oxide-hydroxide corrosion products), this partially acted as a physical barrier between the metal and the environment, controlling the progress of the corrosion in its own way. Thus, the shape of the Nyquist diagrams changed, presenting an incomplete depressed semicircle at high frequencies (100 kHz–100 mHz), followed by semi-linear diffusion impedance at the low-frequency domain (10–100 mHz). However, Nyquist diagrams of the carbon steel (Figure 8b) were in the domain of ≈25 times lower values of Z′ and Z″ compared to those of stainless steel (Figure 8a).

Figure 9 presents the Bode diagrams of stainless steel SS430 (a,b) and carbon steel B450C (c,d). The impedance module of stainless steel |Z| (Figure 9a) showed an increase of ≈100 kΩ cm^2^ (at 10 mHz) after 30 days of exposure to the JLSC1 cement extract. Meanwhile, the phase angle (θ) was kept until 7 days at ≈−75° (Figure 9b) and was ascribed to the capacitive nature of the formed passive layer (composed mainly from Cr_2_O_3_, in the addition of Fe_2_O_3_ and FeO oxides), which was able to accumulate electric charges and block the diffusion of aggressive species, such as oxygen and chloride ions [42,67,68]. At a later period, the phase angle diminished to ≈−70° when the localized attack (pitting) occurred due to the chloride ions, which penetrated through the passive layer.

On the other hand, the impedance module |Z| of carbon steel B450C (Figure 9c) decreased by approximately two times (≈18 kΩ cm^2^) at 30 days of exposure to the JLSC1 cement extract due to the loss of the passive state, as the pH moved to lower alkaline values. The corresponding phase angle values (Figure 9d) stabilized at θ ≈ −35°, although at 10 Hz, the phase angle reached θ ≈ −60°. The lower phase angle confirmed the loss over time of the passive state. This change in the value of θ indicated that the progress in the corrosion process was more complex on the carbon steel surface than that on the stainless steel SS430. To quantify the EIS data, two equivalent circuits were used (Figure 10), where the double-layer capacitance of the interface was replaced with the constant-phase elements CPE1 and CPE2 [69,70,71,72].

The complexity of the B450C carbon–steel interface was simulated with two constant-phase elements (Figure 10b): CPE1 capacitance (high-frequency time constant) of the formed corrosion layer (Fe_2_O_3_/FeOOH) and its defects, while the CPE2 capacitance (low-frequency time constant) was associated with the active corrosion sites on the surface. The electrochemical corrosion activity at the passive stainless steel interface was presented with the simplified Randles circuit (Figure 10a), with only one time constant (CPE1) and the charge transfer resistance (Rct) [49,73,74]. On the other hand, CPE1 could be associated with the Rcp resistance of the formed corrosion layer (to cracking propagation). The Rs is the solution resistance at the electrode/electrolyte interface (which can change with the pH and ionic composition).

Table 8 resumes the fitting parameters obtained from the EIS measurements. The exponential factor of CPE ranged from n = 0.5 to n = 0.9 (for an ideal capacitor, n = 1), the values of which were associated with the properties of constant-phase elements. For stainless steel, n_2_ ≈ 0.88 (CPE2) was relatively constant for up to 30 days, confirming the capacitive nature of the formed passive layer, while for carbon steel, the exponential factor was around n ≈ 0.70 (CPE2), based on changes to the composition of the formed corrosion layer. The fit c^2^ (10^−4^) was good in most cases.

The polarization resistance (Rp) of steel was used as an indicator of the stability of formed films on the steel surface, and this was calculated by the sum of Rcp and Rct (Equation (1)) [66]:(1)$$R_{p}=R_{\mathrm{cp}}+R_{\mathrm{ct}}$$

For the calculation of the *d* thickness, the CPE2 values were used and transformed into the corresponding capacitance values, according to the Brug formula (Equation (2)) [75]. The thickness was calculated from Equation (3) [76,77], where *ε_0_* is the vacuum permittivity (8.85 × 10^−14^ F cm^−1^), and ε is the dielectric constant of the passive layer, which could be assumed to be 15.6 for stainless steel [76,78]
(2)$$C={\mathrm{CPE}\;}^{\frac{1}{n}}{\left(\frac{R_{s}R_{\mathrm{ct}}}{R_{s}+{\;R}_{\mathrm{ct}}}\right)}^{\frac{1\;-\;n}{n}}$$
(3)$$d=\frac{\varepsilon \varepsilon_{0}A}{C}$$

Figure 11 compares the evolution of the passive layer thickness (Figure 11a) and Rp values (Figure 11b) of SS430 and B450C when exposed to the JLSC1 cement extract solution. These characteristics parameters were compared to those previously reported for the same steels exposed to the super sulfated (SS1) cement extract [52] and Portland cement (PC) extract solution [40].

In the JLSC1 cement extract solution, stainless steel SS430 showed an increment in the passive layer thickness and reached ≈0.83 nm at 30 days, while that of the carbon steel B450C was ≈0.1 nm on the first day and tended to disappear (Figure 11a). The SS430 passive layer thickness formed in the SS1 cement extract was quite like that in JLSC1 for up to 21 days of exposure, although at the end of 30 days, it was 0.2 nm lower in SS1. On the other hand, the passive layer on the carbon steel B450C showed a tendency to increase and reach ≈0.3 nm after 30 days of exposure to SS1. The greater values of the passive layer thickness formed on SS430 and B450C steels when exposed to the PC extract solution (Figure 11a) are ascribed to the more alkaline pH of that solution because of the higher content of Ca^2+^ in the absence of chloride ions, which could be closely related to the greater Rp values of both steels (Figure 11b), compared to those obtained for the super sulfated (SS1) and sodium silicate modified limestone-Portland cement extract (JLSC1).

## 4. Conclusions

The corrosion activity of low-chromium ferritic SS430 stainless steel and carbon steel B450C was studied during their exposure for up to 30 days compared to the sodium silicate-modified limestone-Portland cement (JLSC1) extract solution.

(1)At 24 h after the exposure of steels, the initial pH of the JLSC1 cement extract solution (12.60) tended to have lower alkaline values and maintained an average of 9.60 to the end of the immersion test. The decrease in pH could be associated with the dissolution of the CO_2_ from the air environment, forming the carbonic acid (H_2_CO_3_), which dissociated into hydrogen ions (H^+^), promoting the acidification of the cement extract solution, while the bicarbonate aggressive ions (HCO_3_^−^) could stimulate the release of Fe^2+^ and the formation of hydroxides as corrosion products.(2)As a result of the change in pH, the free corrosion potential (OCP) of B450C tended to produce more negative values, as the steel surface lost its possibility to form a passive layer, entering an active corrosion state, while the OCP of the stainless steel SS430 shifted to more positive values, indicating the development of a more protective passive layer along with time.(3)The SEM-EDS and XPS analysis suggested that at 30 days of exposure, the corrosion layer formed on the carbon steel B450C was composed of FeOOH and FeO in similar proportions. The main component of the passive layer on the SS430 surface was Cr_2_O_3_, in the presence of FeO, Fe_2_O_3,_ and Cr(OH)_3_ as a corrosion product.(4)As the cement extract solution contained sulfates, their precipitation was observed on the SS430 surface, which accumulated at the metal–solution interface and could favor the self-repassivation of the steel surface, the passive layer of which could no longer protect against the chloride ion attacks.(5)After the removal of corrosion layers formed on the steel surfaces at the end of the immersion test, the stainless surface of SS430 presented isolated pits due to the content of chloride ions in the JLSC1 extract solution, while on the carbon steel B450C surface, the localized corrosion attacks were more significant and extended in the area.(6)Two equivalent electrical circuits were used for the quantitative analysis of EIS (Nyquist and Bode diagrams) to characterize the corrosion activity of the studied steels at the metal–electrolyte interface. The calculated polarization resistance Rp for stainless steel SS430 was ≈2500 kΩcm^2^ at the end of the immersion tens, and the passive film thickness on the surface was ≈0.82 nm. In the meantime, the Rp of carbon steel B450C was ≈32 times lower in magnitude, and its initial value for the passive layer thickness of ≈0.1 nm tended to disappear.(7)The greater values of the passive layer thickness formed on SS430 and B450C steels during their exposure to the PC extract solution (previously reported) were ascribed to the more alkaline pH of that extract solution due to the higher content of Ca^2+^ (in the absence of chloride ions), which was closely related to the greater Rp values of both steels compared to those obtained for the super sulfated (SS1) and sodium silicate modified limestone-Portland cement extract (JLSC1).(8)The reported results indicated that the change in time of pH and the free corrosion potential (OCP) values were decisively dependent on the cement composition and that of the ions’ presence in the extract solution.

## Figures and Tables

**Figure 1 materials-16-05066-f001:**
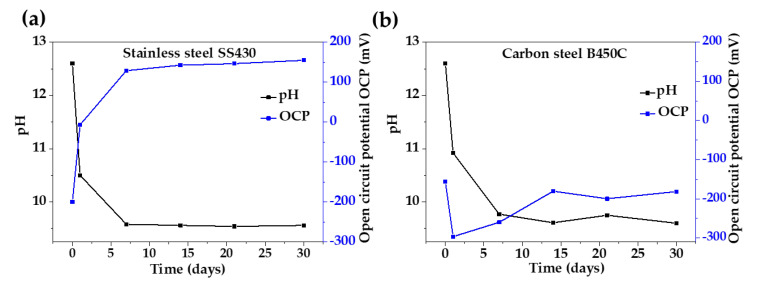
Change in time of pH of JLSC1-CE solution and corrosion potential (OCP) vs. SHE average values during the exposure of (**a**) Stainless steel SS430 and (**b**) Carbon steel B450C.

**Figure 2 materials-16-05066-f002:**
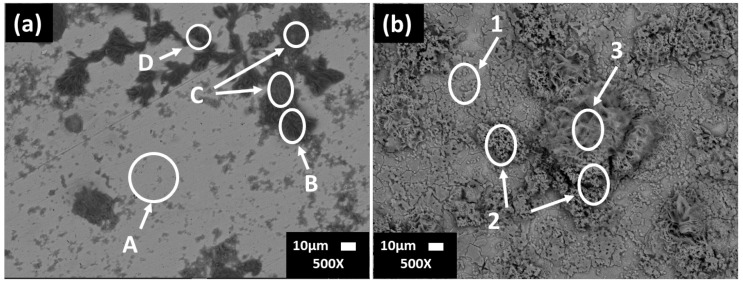
SEM images of carbon steel B450C after 7 (**a**) and 30 days (**b**) of exposure to the JLSC1 cement extract solution.

**Figure 3 materials-16-05066-f003:**
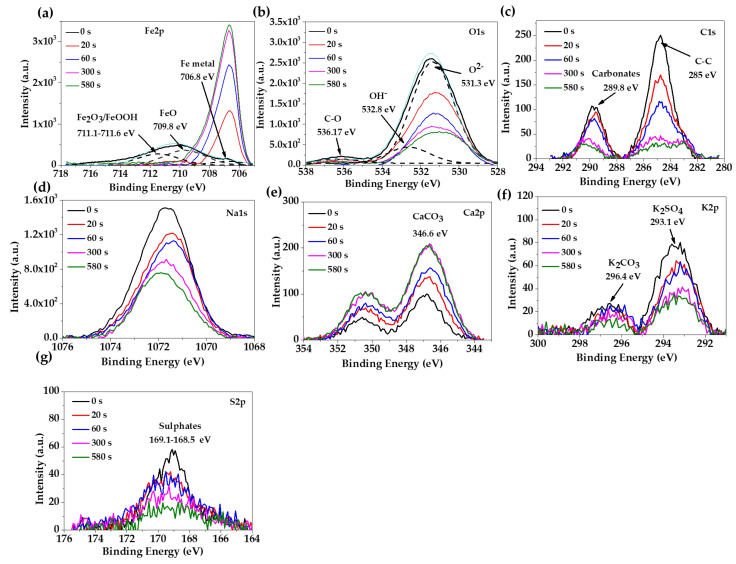
Overview of X-ray photoelectron spectroscopy (XPS) spectra, after sputtering of the metal surface for different times, acquired from carbon steel B450C exposed to the JLSC1 cement extract for 7 days: (**a**) Fe2p; (**b**) O1s; (**c**) C1s; (**d**) Na1s; (**e**) Ca2p; (**f**) K2p; (**g**) S2p.

**Figure 4 materials-16-05066-f004:**
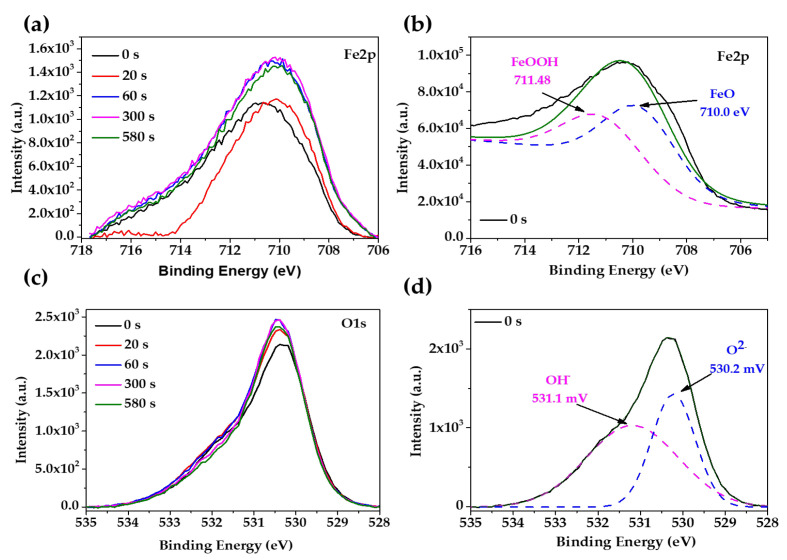
Overview of X-ray photoelectron spectroscopy (XPS) spectra, after sputtering the metal surface for different times, acquired from carbon steel B450C exposed to the JLSC1 cement extract for 30 days: (**a**) Fe 2p; (**b**) deconvolution of Fe2p; (**c**) O1s; (**d**) devolution of O1s.

**Figure 5 materials-16-05066-f005:**
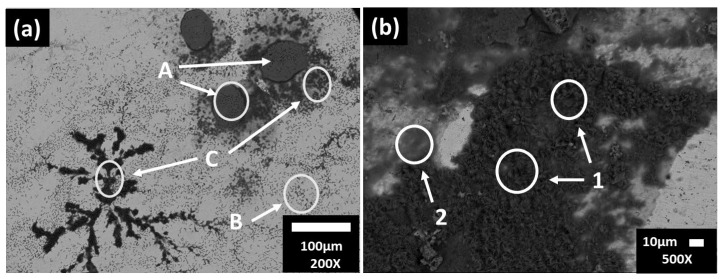
SEM images of stainless steel SS430 after 7 (**a**) and 30 (**b**) days of exposure to the JLSC1 cement extract solution.

**Figure 6 materials-16-05066-f006:**
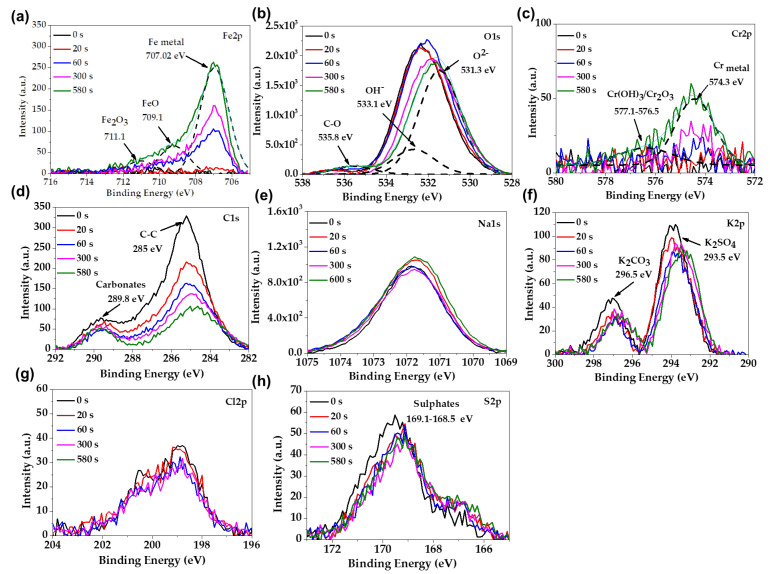
Overview of X-ray photoelectron spectroscopy (XPS) spectra, after sputtering the metal surface for different times, acquired from SS430 steel exposed to the JLSC1 cement extract for 30 days: (**a**) Fe2p; (**b**) O1s; (**c**) Cr2p; (**d**) C1s; (**e**) Na1s; (**f**) K2p; (**g**) Cl2p; (**h**) S2p.

**Figure 7 materials-16-05066-f007:**
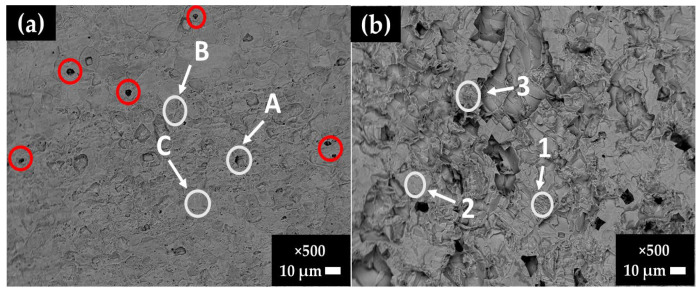
SEM images of (**a**) SS430 and (**b**) B450C steel surfaces after the removal of the layers formed during their exposure to the JLSC1 cement extract solution for 30 days.

**Figure 8 materials-16-05066-f008:**
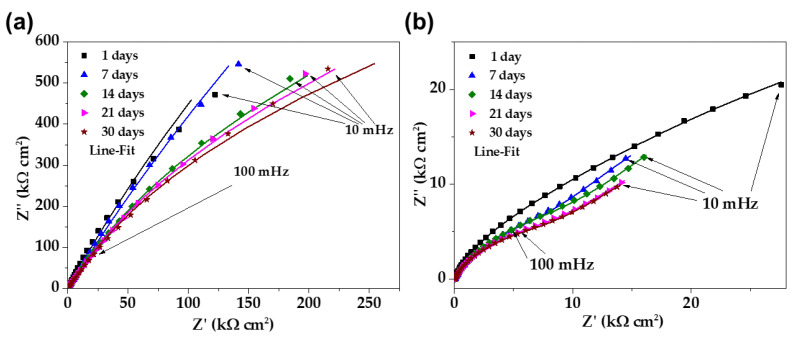
EIS Nyquist diagrams with a respective fitting line for stainless steel SS430 (**a**) and carbon steel B450C when (**b**) exposed to the JLSC1 cement extract up to 30 days.

**Figure 9 materials-16-05066-f009:**
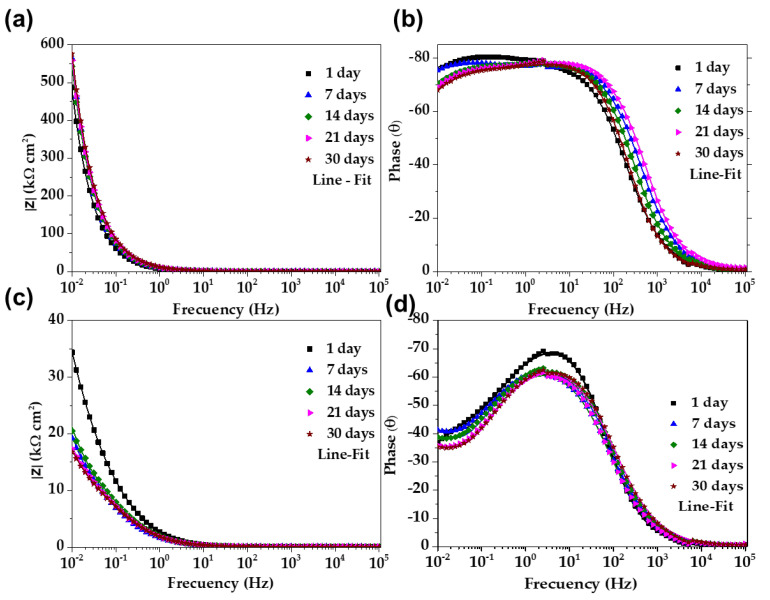
EIS Bode diagrams with the respective fitting line for SS430 (**a**,**b**) and carbon steel B450C (**c**,**d**) after different times of exposure to the JLSC1 cement extract.

**Figure 10 materials-16-05066-f010:**
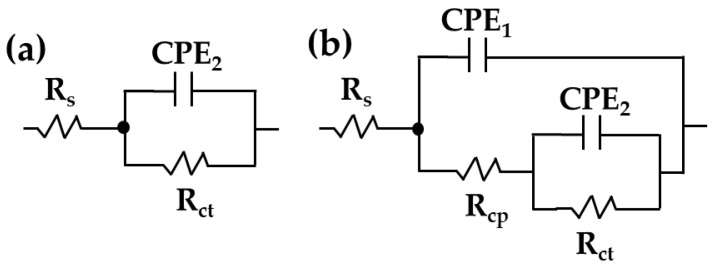
Equivalent circuits proposed for SS430 stainless steel (**a**) and B450C carbon steel (**b**) when exposed to the JLSC1 cement extract solution.

**Figure 11 materials-16-05066-f011:**
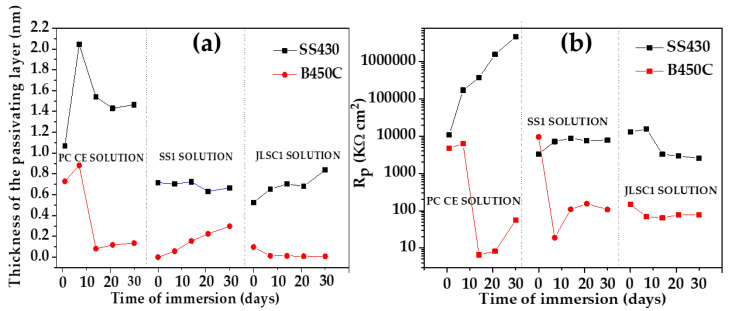
Evolution of the average passive-layer thickness (**a**) and Rp values (**b**) during the immersion of SS430 and carbon steel B450C to the sodium silicate modified limestone-Portland cement extract (JLSC1) extract cement (this study), super sulfated cement extract (SS1) [52] and Portland [40] cement (PC) extract solutions.

**Table 1 materials-16-05066-t001:** Compositions (wt.%) of SS430 ferritic steel and martensitic B450C carbon steel, according to the manufacturers.

Element (wt.%)	C	Cr	N	Cu	P	S	Fe
SS430	0.25	16.2	-	-	-	-	Balance
B450C	0.22	-	0.01	0.80	0.05	0.05	Balance

**Table 2 materials-16-05066-t002:** Oxide composition of the JLSC1 sodium silicate modified limestone-PC.

JLSC1	SiO_2_	Al_2_O_3_	Fe_2_O_3_	CaO	MgO	SO_3_	K_2_O	Na_2_O	LOI
**g**	136.92	10.30	6.67	285.16	3.66	6.76	2.30	19.49	128.82
**wt.%**	22.80	1.72	1.11	47.53	0.61	1.13	0.38	3.25	21.47

**Table 3 materials-16-05066-t003:** Ion chemical composition in mg/L of the JLSC1 extract solution obtained by absorption spectrometry (AA), atomic emission by the plasma (ICP), and the ion selective (Cl-ion).

Element (mg/L)	Li	K^+^	Na^+^	Al^3+^	Ca^2+^	Si	SO_4_^2−^	Sr	Cl^−^	OH^−^
JLSC1	0.099	1668.4	2700	2.555	0.505	34.96	1132.0	1.005	88	676.78

[OH] was calculated from the pH measurement.

**Table 4 materials-16-05066-t004:** EDS surface average analysis (wt.%) of the B450C carbon steel areas of interest as marked on the SEM images of Figure 2.

Days/wt.%		Fe	O	Na	Ca	S	K	Si	C	Mn	Cl
7	A	88.71	3.12	0.64	0.42	-	-	0.76	5.00	0.74	-
	B	2.10	51.42	22.57	1.11	0.15	1.90	0.19	15.58	-	4.48
	C	9.81	42.47	10.41	1.99	12.19	17.04	0.39	5.72	-	-
	D	57.37	15.23	20.68	-	-	0.59	0.27	4.08	-	1.78
30	1	60.28	36.58	-	-	0.40	-	0.48	2.25	-	-
	2	55.35	40.35	-	-	0.40	-	-	2.86	0.44	0.47
	3	51.30	42.31	-	-	0.88	-	-	4.49	-	0.51

**Table 5 materials-16-05066-t005:** EDS surface average analysis (wt.%) of SS430 areas of interest as marked on the SEM images of Figure 5.

Days/ wt.%		Fe	Cr	O	Na	S	K	Si	C	Cl
7	A	1.21	-	36.20	8.28	22.87	31.19	0.24	-	-
	B	64.28	14.00	6.63	3.53	-	0.24	0.67	10.74	-
	C	46.93	9.77	14.62	21.69	4.77	0.33	-	1.88	-
30	1	4.08	1.23	52.72	22.77	0.33	0.43	1.06	17.02	0.35
	2	32.57	7.58	25.22	5.78	-	1.47	9.10	15.31	0.28

**Table 6 materials-16-05066-t006:** EDS surface average analysis (wt.%) of the SS430 areas of interest as marked on the SEM image of Figure 7a.

wt.%		Fe	Cr	V	Si	C
SS430	A	58.12	40.09	1.79	-	-
	B	50.26	43.51	2.10	0.46	3.67
	C	74.29	15.61	-	-	10.11

**Table 7 materials-16-05066-t007:** EDS surface average analysis (wt.%) of the B450C areas of interest as marked on the SEM image of Figure 7b.

wt.%		Fe	Mn	C	O
B450C	1	89.92	0.72	6.05	3.31
	2	99.39	0.61	-	-
	3	73.98	0.70	9.56	15.76

**Table 8 materials-16-05066-t008:** Fitting parameters obtained from EIS measurements for SS430 and carbon steel B450C when exposed to the JLSC1 cement extract up to 30 days.

Days	R_sol_ kΩcm^2^	R_cp_ kΩcm^2^	CPE_1_ μSs^n^cm^−2^	n_1_	R_ct_ kΩcm^2^	CPE_2_ μSs^n^cm^−2^	n_2_	R_p_ kΩcm^2^	c^2^ 10^−4^
1	0.07	-	-	-	13,000	24.29	0.88	13,000	4.88
7	0.05	-	-	-	15,630	19.67	0.86	15,630	3.84
14	0.07				3273	19.06	0.87	3273	3.11
21	0.04				2962	18.24	0.87	2965	5.78
30	0.10				**2542**	17.16	0.87	**2542**	11.09
1	0.08	8.32	66.99	0.86	138.70	94.78	0.49	147.02	0.69
7	0.06	15.80	146.50	0.75	54.10	481.30	0.79	69.90	1.26
14	0.06	15.97	122.00	0.76	48.14	483.00	0.79	64.11	2.98
21	0.07	12.57	124.90	0.75	65.54	468.40	0.67	78.11	2.22
30	0.06	11.59	122.10	0.76	**65.97**	457.00	0.65	**77.56**	2.05

## Data Availability

The data are available upon request from the corresponding author.

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
