# Peer review of "Corrosion Activity of Stainless Steel SS430 and Carbon Steel B450C in a Sodium Silicate Modified Limestone-Portland Cement Extract"

_materials, 2023, doi:10.3390/ma16145066_

Round 1

Reviewer 1 Report

This is a carefully done study and the findings are of considerable interest. A few minor revisions are list below.

1.       In table.1, is there any other elements in the steel, such as Ni, Mo, Mn, Si?

2.       What is the microstructure phase of the B450C andSS430 steel?

3.       All the test data should be added error bars.

4.       The data in Table.4 should be change as a figure.

5.       What are the corrosion products and their chemical reaction?

6.       The conclusion section should be divided several parts, e.g. (1)XXX; (2)XXX; (3)XXX.

Author Response

Reviewer 1: This is a carefully done study and the findings are of considerable interest. A few minor revisions are listed below.

  1. In table.1, are there any other elements in the steel, such as Ni, Mo, Mn, Si?

Answer: According to the manufacturers for stainless steel SS430 (Outokumpu Handbook of Stainless Steel 2017 edition.pdf) it was not reported other elements. Below Table 1. It is mentioned that elements such as N, V and Si were detected on the surface of the SS430, according to the reported references, although these elements are not considered from the manufacturers. For carbon steel in our cited previous study the Mn was detected through SEM-EDS, besides the N, Cu, P, and S reported for the manufacturer (REINFORCING STEEL BARS B450C (pittini.it))

  1. What is the microstructure phase of the B450C and SS430 steel?

Answer: The microstructure of the stainless steel SS430 is ferrite and this fact is mentioned in the materials section. That of the carbon steel B450C is martensite and it will be added in the section 2.1. Steel Samples and Surface Characterization.

  1. All the test data should be added error bars.

Answer: We have not added the error bars because average values (triplicated samples) were used for elaboration of tables and figures. For this reason, the word “average” was included in the captions of tables and figures. 

  1. The data in Table.4 should be changed as a figure.

Answer:The Table 4 was transformed in to the new Figure 1. 

  1. What are the corrosion products and their chemical reaction?

Answer: The corrosion product Cr(OH)3 formed on the stainless steel SS430 is considered as a product of Cr hydrolysis, according to the literature.

Xu, L.; Wang, B.; Zhu, J.; Li, W.; Zheng, Z. Effect of Cr content on the corrosion performance of low Cr-alloy steel in a CO2 environment. Appl. Surf. Sci. 2016, 379, 39-46. https://doi.org/10.1016/j.apsusc.2016.04.049

On the carbon steel B450C surface, XPS analysis suggested that the corrosion layer is mainly formed by FeO, Fe2O3 and FeOOH. Due to the presence of Cl ion in the cement extract, the formation of the high soluble FeCl3 corrosion product is promoted. However, this is transformed latter to Fe hydroxide and oxides.

Adewale Adewumi, A.; Maslehuddin, M.; Al-Dulaijan, S.; Shamen, M. Corrosion behavior of carbon steel and corrosion resistant steel under elevated temperature and chloride concentration in simulated concrete pore solution. Eur. J. Environ. Civ. Eng. 2021, 25, 452-467. https://doi.org/10.1080/19648189.2018.1531270

  1. The conclusion section should be divided several parts, e.g. (1)XXX; (2)XXX; (3)XXX.

Answer: We considered your suggestion.  

Reviewer 2 Report

In this paper the corrosion activity of low-chromium ferritic SS430 stainless steel and carbon steel B450C was studied during their exposure up to 30 days to sodium silicate modified limestone-Portland cement (JLSC1) extract solution,  to simulate the environment at the steel-concrete pore interface.

The pH of the CE solution and the change in time of the steel free corrosion potential (OCP) were monitored during the immersion test.

SEM-EDS and XPS analysis were applied to study the formation of  the corrosion layer  on the carbon steel B450C and  on the SS430 surfaces.

Two equivalent electrical circuits were used  to characterize the corrosion activity of the studied steels at the metal-electrolyte interface.

The paper is clear and well written. Both experimental results and modelling are of interest to evaluate replacements of PC without compromising the steel mechanical properties.

Some remarks

The abstract is too long and does not reflects adequately the content of the paper. 

The content of the paper a the end of the introduction is not complete.

Author Response

Reviewer 2: In this paper the corrosion activity of low-chromium ferritic SS430 stainless steel and carbon steel B450C was studied during their exposure up to 30 days to sodium silicate modified limestone-Portland cement (JLSC1) extract solution, to simulate the environment at the steel-concrete pore interface. The pH of the CE solution and the change in time of the steel free corrosion potential (OCP) were monitored during the immersion test. SEM-EDS and XPS analysis were applied to study the formation of  the corrosion layer  on the carbon steel B450C and  on the SS430 surfaces. Two equivalent electrical circuits were used  to characterize the corrosion activity of the studied steels at the metal-electrolyte interface. The paper is clear and well written. Both experimental results and modelling are of interest to evaluate replacements of PC without compromising the steel mechanical properties.

            Some remarks:

The abstract is too long and does not reflects adequately the content of the paper. 

The content of the paper at the end of the introduction is not complete.

Answer: After the revision of the suggested, the author consider that the abstract and the indroducction are well presented, giving the information, which will help the readers to understand the content of our article.

Reviewer 3 Report

a separatred discussion part should be added before the conclusion part, to leave a global synthesis about all the different results and to explain the results evolution.

the results presentaiton should be improved:

- in table 4 and in figure 10, it is necessary to add the measurement precision/error;

- in caption of table 5, it is necessary to refer the analysis zone to the figure 1; it is important to precise the analysis precison/error;

- in caption of table 6, it is necessary to refer the analysis zone to the figure 4; it is important to precise the analysis precison/error;

-in caption of table 7 and 8, it is necessary to refer the analysis zone to the figure 6; it is important to precise the analysis precison/error;

Author Response

Reviewer 3: a separatred discussion part should be added before the conclusion part, to leave a global synthesis about all the different results and to explain the results evolution. the results presentaiton should be improved.

  1. In table 4 and in figure 10, it is necessary to add the measurement precision/error.

Answer: We have not added the error bars because average values (triplicated samples) were used for elaboration of tables and figures. For this reason, the word “average” was included in the captions of tables and figures. The Table 4 was transformed in to the new Figure 1. 

  1. In caption of table 5, it is necessary to refer the analysis zone to the figure 1; it is important to precise the analysis precison/error.

Answer: The caption of table 4 (ex Table 5) was changed:

“Table 4.  EDS surface average analysis (wt.%) of the B450C carbon steel areas of interest as marked on the SEM images of Figure 2”.

Note: according to the suggestion of one of the reviewer, the table 4 was transformed in to new figure.

  1. In caption of table 6, it is necessary to refer the analysis zone to the figure 4; it is important to precise the analysis precison/error.

Answer: The caption of table 5 (ex Table 6) was changed

“Table 5. EDS surface average analysis (wt.%) of the SS430 areas of interest as marked on the SEM images of Figure 5”.

  1. In caption of table 7 and 8, it is necessary to refer the analysis zone to the figure 6; it is important to precise the analysis precison/error.

Answer: The captions of Table 6 and 7 (ex Tables 7 and 8) was also changed, as suggested.

“Table 6. EDS surface average analysis (wt.%) of the SS430 areas of interest as marked on the SEM 335 image of Figure 7a.

“Table 7. EDS surface average analysis (wt.%) of the B450C areas of interest as marked on the SEM image of Figure 7b.  

Round 2

Reviewer 3 Report

in the revised version of the manuscript, authot has made several corrections including some remarks from reviewers. 

But the most important change to value de studied results has not be done, this disucssion is absolutely necessary  "a separatred discussion part should be added before the conclusion part, to leave a global synthesis about all the different results and to explain the results evolution".

Author Response

Dear reviewer No.3,

you mention that “In the revised version of the manuscript, author has made several corrections including some remarks from reviewers”. 

Answer: The several changes made, that your mention in your report, are not essential and they do not change our results and their discussion. For example: 

  • The Table 4 was transformed into the new Figure 1, according to the suggestion of the Reviewer 1. This is not an essential change, but the number of the rest of Figures, including their mention in the text, was reorganized.
  • The data in Figures and Tables (EDS) represent average values (of the triplicate tested samples, as it mentioned in the experimental part) and for this reason their error bars were not included. However, to be clear for the readers, the word “average values” was added to the title of Figures and Tables. (This is not an essential change, which affects the results (values) reported in our article).

Your mention that is absolutely necessary  "a separatred discussion part should be added before the conclusion part, to leave a global synthesis about all the different results and to explain the results evolution".

Answer: Our experience is to discuss the results immediately after their presentation in each part of the article. This helps to relate the results reported in each part with the next data  and next data  ….until the end of the final results. Making a resume of the discussion (repeat even in resumed form) is not necessary. However, the conclusions are exactly the resumed form of the discussed results before in each part ot the article, giving the most important, not published before results.

We consider that our abstract (in the limit of 200 words) has been well presented.

Please, consider our experience and knowledge in electrochemistry and corrosion, please. The Mexcan researchers and their teams are working well.

The authors appreciate all Reviewers comments , suggestions annd time very much.

On behalf of all authors:

 Prof. Dr. Lucien Veleva

Member of MDPI Editorial Board of Journals

Doctor Honoris Causa in Corrosion (2011)

“Francis Laque” Award of ASTM (G01, 2012)

NACE Award for International Distinguished Career (2013)

Service Recognition Award, ASTM E07, Nondestructive Testing (2017) 

Mexican Society of Electrochemistry (section of ECS) – National Award of 2020

CINVESTAV-IPN, Applied Physics Department,

Laboratory of Physical Chemistry

Merida, Yucatan State, Mexico